# Strategic Thinking and Its Role in Accelerating the Transition from the Linear to the Circular Economic Model—Case Study of the Agri-Food Sector in the Sibiu Depression Microregion, Romania

**Romulus Iagăru, Anca Șipoș** * **and Pompilica Iagăru**

Faculty of Agricultural Sciences, Food Industry and Environmental Protection, Lucian Blaga University of Sibiu, 7-9 dr. Ion Ratiu Street, 550012 Sibiu, Romania
* Correspondence: anca.sipos@ulbsibiu.ro

**Abstract:** Our research provides solutions to alleviate the economic problems currently plaguing our planet that are responsible for the decline of its ecological systems. Our motivation is the need to identify elements that will encourage and accelerate the transition from a linear to a circular economic model, raising awareness of the limited nature of resources and the major pressures exerted by climate change and population growth. Our paper highlights the implications of strategic thinking, i.e., strategic management, in the development and promotion of the circular economy, including the concept of sustainability, in the agri-food sector. We propose strategic options based on information from our secondary analysis of statistical data and relevant literature, e.g., from PESTEL, SWOT, and DPSIR diagnostic models, for integrating resource flows into circular processes, which are meant to reduce resource consumption and minimise waste. Our paper elaborates on an integrated and dynamic model for the transition from a linear to a circular economic model; furthermore, we perform further research to create appropriate frameworks for elaborating on and implementing the most relevant policy options to accelerate this transition process.

**Keywords:** accelerating the transition; circular economy; climate change; strategic management; strategic options; resources

## 1. Introduction

Strategic thinking integrates many one-dimensional thinking models to provide an understanding of the ways that societies can build toward investing in sustainable development goals in the form of developing strategic objectives [1].

Strategic options are the result of strategic decisions (from top management) based on strategic thinking, and they aim to achieve strategic objectives in development.

Achieving the strategic objectives assumed by the company or enterprise implies a transition from a present to a future state—the agri-food sector's transition from a linear to a circular economy. However, achieving the transition is not simple and involves many factors, and choosing the most appropriate one in a given context is based on rational analysis processes, which involve many decisions and activities that require resources and therefore generate costs.

Our world is characterised by complexity, which is why the process of choosing the most appropriate option for achieving objectives, i.e., transitioning to a future state, is also complex and requires taking into account an environment defined by rapid and difficult-to-anticipate changes.

It is imperative to develop strategic thinking and provide adequate managerial training to set strategic objectives and develop tactics to achieve them [1].

The general context in which humanity must establish relevant objectives and strategies for the transition to the circular economy is dominated by a summation of negative

effects as a result of the linear economic model. This led to the "depletion of the natural capital on the planet considered our home" [2]; the decrease and degradation of natural resources; air and water pollution; and the decline of natural ecosystems [3,4]. Therefore, the concerns of specialists regarding the acceleration of the transition to the circular economic model are justified and have been intensifying since the last decade. There is a unanimous recognition that the economy developed on "take, make, consume and dispose" is not sustainable, harms the health of the environment, and contributes to the decline of natural ecosystems [5–9]. A new orientation is needed, focused on the sustainability of ecosystems based on strategic thinking, whose economic goal is to maintain as much as possible the value of products, materials, and resources and to significantly reduce the amount of waste. Thus, the circular economy manages to attract more and more followers who energise the approaches of specialists regarding the definition of the concept, far from reaching a unanimously accepted definition. Most specialists revolve around the following key concepts: sustainable development, the framework of the 4Rs (Reduce, Reuse, Recycle, Recover), the systemic approach (micro, meso, macro), the waste hierarchy [10]. The need to promote economic, social, environmental, and technological elements in the process of development is also obvious [11]. At the same time, there is a need to promote change in agricultural technology and the food industry to stop the destabilization of ecosystems and climate change [12]. The successful implementation of the circular economy concept needs political and legislative support, which is why it is becoming an omnipresent topic on the political agenda at the level of the European Union [13]. Next, we can see the generation of instruments that support the transition to the circular economy model, such as the circular economy package [14], the economic growth strategy adopted by the European Commission (EC), the European Green Deal [15] (EC, 2019), the law for the promotion of the circular economy in China [16], etc.

Scientific forums also express support for promoting a circular economy through the conduct of specific research and the dissemination of results through publications in the form of case studies, reviews, scientific reports, and articles, converging towards the following:

- An enterprise-level adoption of strategies aimed at achieving sustainable production based on the 10Rs concept and Industry 4.0 technology [17];
- Judicious allocation of resources and the use of the Industry 4.0 concept in agriculture [18];
- The valorisation of waste (sludge) from drinking water treatment plants [19];
- Analyzing the perceptions of the circular economy in Romanian SMEs in accordance with the six ReSOLVE framework actions, half of which are correlated in terms of value creation (regeneration, optimisation, and exchange) [20];
- Implementing the 3R principles (reduce, reuse, and recycle) and efficiently managing natural resources [21];
- Increasing education levels [22];
- Minimising the use of natural resources in production processes [23,24];
- Optimising resource consumption by implementing waste management legislation [25];
- Reusing products that still have operational functionality [26];
- Extending the useful life of products [27].

The succinct image of what is presented is successively captured in Figure 1, where ensuring the sustainability of the economy is attributed to the implementation of the circular economy by accelerating the transition to it and promoting its key elements at the level of the economic, social, environmental, and technological pillars.

The circular economy will succeed in achieving its ideals if businesses choose to embrace it strategically and operationally and if governments develop consistent sets of rules. All this becomes possible if consumer behaviour also changes [28].

Production models based on the 10R philosophy are applicable to the agri-food sector, helping farmers and manufacturers to produce more by using less. Much of what we now consider waste in the agri-food sector can be processed and used as animal fodder, in bioenergy production, or as fertiliser for agricultural land [29].

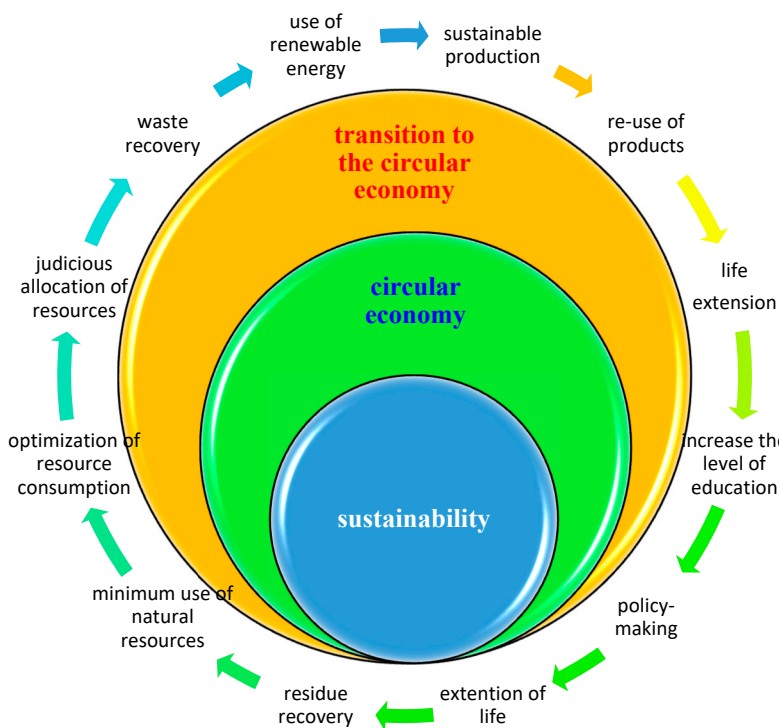

**Figure 1.** Sustainability of the economy based on the circular economy.

Farmers' initiatives to promote integrated farming systems and close as many cycles as possible to reduce the amount of waste should be commended. This is possible by promoting innovation in production and consumption processes, such as adopting artificial intelligence in production processes, e.g., using wireless sensor networks and UAVs for data collection and software for data processing, IoT technology for interpreting results and choosing optimal decision options, and GPS-enabled machinery for input management and technology deployment. We must also promote development in consumption processes through education, e.g., by changing consumption behaviours and developing appropriate tools (including legislation) for transitioning to a circular economy model [30].

This can create the conditions for developing new business models based on optimising the use and reuse of resources, such as the transformation of mangos, apples, cacti, or even coffee grounds into fashion items; the creation of new production flows for vegetables rejected on aesthetic grounds by retailers; treating coffee grounds as a raw material for beech trout–*Pleurotus ostreatus Jacq* (a fungus species cultivated on an international scale); managing and reducing waste efficiently by composting waste from food businesses using composting stations and then using the resulting fertiliser as a substrate for herbs used in product preparation [31].

The EU-level agricultural sector has intense concerns about the transition to circular agriculture. The transition requires partnership and innovation, which creates an appropriate framework for sharing knowledge and experiences towards redesigning the current supply chain, namely raw material supply, production, processing, packaging, storage, and distribution [32]—its current motto is "from farm to fork".

The economic and social context is favorable to the development of the circular economy, as can be seen from the interest of researchers, from scientific publications, and not least from the legislative and financial support for the promotion of this concept. The circular economy in the agri-food sector is of particular importance due to the role it plays in the generation of welfare and social development, as well as in the balance of the environment [33]. Furthermore, an analysis carried out by Hamam et al. regarding circular economy models in the agri-food sector highlights the need to implement cleaner

production models [34]. In agriculture, the circular economy is seen as an economic model that respects the environment and offers emerging business opportunities [35].

## 2. Materials and Methods

We believe that socio-economic reality is best revealed using "in parallel and complementary ways quantitative and qualitative methods to achieve added knowledge" [36]. Thus, we used methods and techniques belonging to strategic management because of their usefulness in obtaining an overview of the transition from the linear to the circular economic model, allowing us to identify critical factors that impact the process. Therefore, we conducted a secondary literature review, identified critical factors and successful initiatives, and applied the PESTEL (political, economic, social, technological, natural, and legislative) and SWOT (strengths, weaknesses, opportunities, and threats) analysis models.

We used our results to set up the DPSIR (driving forces, pressure, state, impact, response) model at a microregional level to establish an integrated and dynamic vision of the agri-food sector's transition from a linear to a circular economic model.

Figure 2 shows the schematic structure of the research.

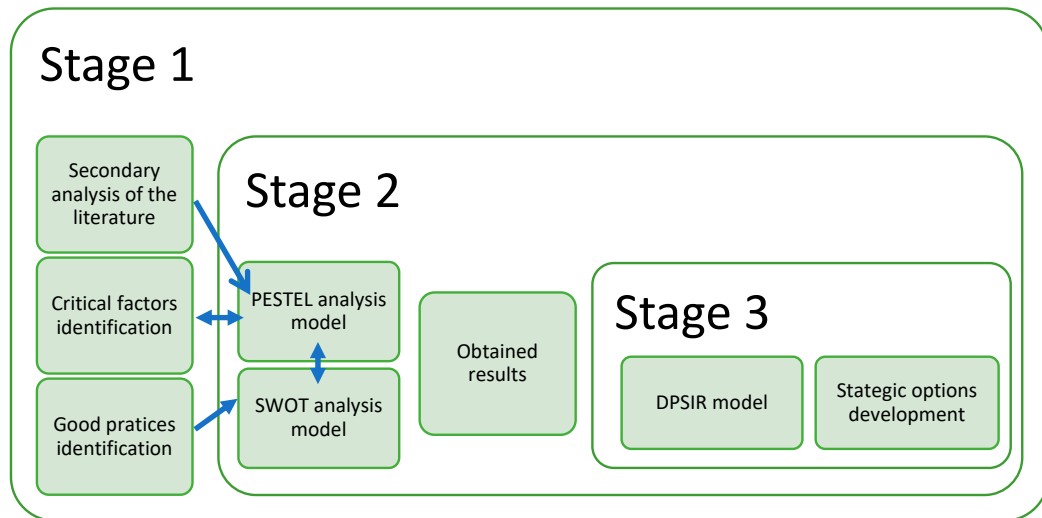

**Figure 2.** Schematic structure of research.

We integrated the methods we adopted in our research into our case study methodology, which we recognized for its ability to adapt to territorial specificity [37]; therefore, we focused on the Sibiu Depression, which is part of the Central Development Region of Romania.

Territorial specificity is relevant for the adoption of development directions, and once the general strategic framework is created through legislative measures developed at the European and national levels, its approach in relation to the agri-food field highlights relevant options for the transition to the circular economy.

In order to achieve the goal, namely to develop relevant strategic options to accelerate the transition process from the linear to the circular economic model, the research was carried out in three stages. The first stage consisted of collecting data and information useful for the research. Concretely, a questionnaire focused on the collection of data and quantitative information at the level of the territorial administrative unit (UAT), and the secondary analysis of statistical data and relevant specialised literature was applied. These led to the shaping of a realistic picture and the identification of critical factors and good practices, taking into account the need to stop the degradation of ecosystems and climate change in accordance with the European Green Deal strategy. To gain more knowledge, participatory observation was also used, a qualitative method of collecting data and information that allowed combining data sets and obtaining answers to comparative questions [38]. The second stage is dedicated to the strategic diagnosis made at the level

of the Sibiu Depression microregion by using the PESTEL and SWOT strategic analysis models. The use of the PESTEL analysis model assumed the grouping of the life framework of the Sibiu Depression into a set of six criteria (political, economic, social, technological, environmental, and legislative), their analysis, and the identification and understanding of the macroeconomic forces with an impact on the transition to the model of a circular economy in the agri-food sector. Specific characteristics of the community or area studied were identified, and an important step was taken to develop relevant strategic options. In addition to the obtained information, the SWOT analysis model was used, with the help of which the specific internal and external characteristics of the studied community were combined. The combination of these characteristics outlined four quadrants to which certain strategic options corresponded. The third stage is dedicated to the development of relevant strategic options for accelerating the transition from a linear to a circular economic model in the agri-food sector. For this, we proceeded to identify the factors and understand the relationship between them and the processes of implementing the circular economy using the DPSIR model. The application of the model led to the highlighting of the relationship between the "driving forces" and the political response [39] and the establishment of an integrated and dynamic vision to accelerate the transition process to the circular economic model for the agri-food sector. Interactions were identified between the analysed components, such as those related to consumer need (driving forces) and its effects on the environment (pressures), under the impact of a certain production and consumption model. This interaction generates the need for change so that, with the help of technologies, the present situation (state) can be overcome and the impact on the environment minimised by obtaining the most relevant answer to the question of what must be completed for the transition to the circular development of the agri-food sector.

The research methodology used is recommended by the utility proven in numerous similar studies, such as: analysis of the area for new businesses in the raw materials market [40]; circular economy strategies, implementation, and integration [41]; qualitative analysis of stakeholders for a regional biogas development [42]; an integrated swot-pestel-ahp sustainability assessment model [43]; etc.

To achieve our goals, we set specific objectives representing proposed stages in our research: to perform a secondary analysis of statistical data and literature relevant to the promotion of the circular economy and its adoption in the agri-food sector; a strategic analysis of the general external and competitive environments in the Sibiu Depression microregion; an elaboration of strategic options relevant to the promotion and acceleration of the transition to a circular economy; and establish an integrated and dynamic vision for the agri-food sector's transition to a circular economic model.

The motivation for choosing the Sibiu Depression microregion as the object of the case study in this research is based on its high habitat and geoproductive potential. The Sibiu Depression is located in Romania, more precisely in the southwest of the Transylvanian Hilly Depression and in the northern part of the Southern Carpathians, and is polarised by the city of Sibiu—a strong urban center in continuous development. A municipality, two towns, 9 communes and 8 belonging villages are part of the Sibiu depression. The agricultural area totals 47,495 ha, of which 58.73% belongs to the categories of pastures and hayfields, and 40.01% belongs to the category of arable use, which highlights the favorability for animal breeding and the development of the food industry. The Sibiu Depression is located on the outskirts of a traditional area recognised for its tradition in animal husbandry, the promotion of traditions, and the obtaining of food products, namely Mărginimea Sibiului. At the same time, the development of agri-food education in the city of Sibiu, both pre-university and university type, justifies the choice of this theme.

The Sibiu Depression is a micro-region with multiple development possibilities, including for the agri-food sector, where the tradition of raising animals and processing agricultural products is recognised by numerous brands (Sibiu salami, telemeaua de Sibiu (Sibiu chees), etc.). This forces the continuation of the development process of the agri-food sector, but under the conditions of environmental protection through the implementation of



the circular economy concept. The research presents the implications of strategic thinking, or strategic management, in the development of strategic options for accelerating the transition to the circular economic model. Their development required the research of a model for collecting data, quantitative, and qualitative information, for strategic diagnosis of the area to highlight specific characteristics, and for combining internal and external characteristics, for an integrated and dynamic vision of the specific decision-making context, respectively, highlighting cyclical cause-effect relationships. This model can be replicated in different areas with specific characteristics.

## 3. Results and Discussion

Our research allowed us to identify the critical factors that could impact the transition from a linear to a circular economic model.

### 3.1. Obtained Results from Using PESTEL Diagnostic Model

We carried out an in-depth analysis of the Sibiu Depression microregion using the PESTEL diagnostic model and focusing on the agri-food sector. Our most relevant results from using the PESTEL diagnostic model are as follows:

Our analysis of the political criterion highlights the 2015 EU-level creation of a legislative framework called the "Circular Economy Package (II)", which is dedicated to stimulating the transition towards a circular economy. It contains legislative proposals aimed at reducing waste and increasing recycling and reuse levels. In 2018, the European Commission published the Circular Economy Mini-Package, which contained the following:

- A European Strategy for Plastics in a Circular Economy [44] and a Monitoring Framework for the Circular Economy [45];
- EU Directive 2018/851 amending Directive 2008/98/EC on waste, which calls on Member States to take measures to reduce food waste at each stage of the supply chain;
- EU Directive 2018/852 amending Directive 94/62/EC on packaging and packaging waste aims, among other things, to increase the share of reusable packaging.

It also proposes an integrated and interlinked approach throughout the lifecycle of products. These documents acknowledged that the transition to a circular economy is dependent on updating national legislation and supporting policymakers and stakeholders in developing specific standards. The role of financing in implementing a circular economy is also crucial, and the post-2020 financing framework offers relevant opportunities for the efficient use of resources and implementing a circular economy.

At the national level, a number of measures exist that support the transition to a circular economy. These measures began in 2008 with the promotion of sustainable consumption and production practices in Romania's National Strategy for Sustainable Development Horizons 2013–2020–2030 [46], e.g., the decoupling of economic growth from environmental degradation. In the field of waste management, the National Waste Management Plan (PNGD) approved by GD no. 942/20.12.2017 [47] promoted a shift from the "landfilling of waste to selective collection and recovery" as an objective of integrated waste management.

Law No. 217/2016 [48] focused on reducing food waste, and implemented a number of legislative acts aimed at transposing European Union directives towards a green economy, such as Law No. 160/2016 [49] (on energy efficiency), Emergency Ordinance No. 24/2017 [50] (promotion of energy production from renewable energy sources), etc.

In the Sibiu Depression, support for sustainable development is coordinated by local public authorities, including the Municipality of Sibiu, through its Local Agenda 21 Programme [51], which supports the rebalancing of socio-economic interests and their impact on the environment.

It is important for politicians to support the transition to a circular economic model to ensure sustainable methods of exploiting natural resources; therefore, they must be called upon to adopt the appropriate political measures to update national legislation and transpose European legislation directives. Further policy measures are needed to improve

productivity, which will also help to increase the quality of production, leading to growth in the area's agri-food sector.

Our analysis of the economic criteria covers all the elements that make up the economic life [52] of the Sibiu Depression's agri-food sector. Specifically, the criteria target a set of indicators specific to economic relations: the dynamics of macroeconomic indicators; the GDP per capita ratio; bank interest rates; levels of inflation, imports, and investment; the cost of utilities; the cost of energy; the population's consumption capacity; etc., according to the PESTEL analysis model.

Agriculture is a well-represented activity in the Sibiu Depression, constituting the main occupation of the inhabitants of the region's rural areas, and the agri-food sector benefits from the land's diversified natural potential (15.5% of the agricultural area of Sibiu County), which must be exploited in a sustainable way. Existing agricultural areas in the Sibiu Depression comprise pastures and meadows (58.73%) and arable land (40%)—other categories constitute a small share of 2% [53]. Land use in the Sibiu Depression is important for the rural population's animal husbandry jobs, constituting a challenge for specialists, stakeholders, and organisations in composing a circular agriculture program.

The Sibiu Depression also processes many agricultural products, and more than a third of the area's municipalities have developed capacities, in addition to the existing processing capacities in urban areas. Additionally, there is particular interest in alternative energy production (hydro, wind, and solar power). At the level of the Sibiu Depression, the aim is to achieve a synergy between the food industry and agricultural production, respectively, and a substantial added value as a result of the completion of the two areas and the intensification of specific activities in conditions where the trend of these activities is an increasing one. Additionally, the two industrial parks located in the studied area (Industrial Park Sibiu-Sura Mică and Industrial Park Sibiu-Selimbăr) contribute to improvements in the business environment; therefore, we identified a number of economic factors that could influence the transition from a linear to a circular economic model, including: inflation, the costs of agri-food products, the cost of technologies, etc.

Our analysis of social factors consisted of addressing specific subcriteria: demography, migration, education, attitudes towards quality and saving, etc. The population of the Sibiu Depression is differentiated both quantitatively and qualitatively, influencing the transition to a circular economy by population number, economic structure, level of education and professional training, etc. Approximately 211,800 people live in the Sibiu Depression, which represents almost 50% of the population of the county of Sibiu. Of these, only 16% live in rural areas, showing an unbalanced population distribution because of the inclusion of the municipality of Sibiu and the towns of Cisnădie and Tălmaciu. The average population density is 179 inhabitants per km$^2$; however, it is 46 inhabitants per km2 in the depression's rural areas. The average emigration rate in the Sibiu Depression is around 2501.05 people, i.e., 11.47%. The population's educational level shows an upward trend with TAU-level differentiations [53].

Our analysis of technological factors generally concerns the extent of changes in the IT field, their frequency and interconnections with other systems, and the capacity of the agri-food sector to assimilate new technologies and move towards the development of technological projects beneficial to a circular economy. Therefore, further studies are needed on the agri-food sector's innovation capacity, including the expenditure required to implement new technologies, automate processes, increase research and development levels, etc. In the Sibiu Depression, there is a relatively high level of expenditure on innovation, which is influenced by the two industrial parks and tourism; therefore, the depression is characterised by good receptivity to innovation and cooperation.

Our environmental analysis regarding strategic thinking underlines the need to carry out economic, social, and cultural activities in close harmony with the specific features of the surrounding environment and under conditions of environmental protection. The Sibiu Depression has an asymmetrical relief with hilly fragmentation and a predominance of alluvial plains (60%). The altitude varies between 380 m and 602 m, and the layout is

in steps, similar to an amphitheater. The Sibiu Depression's climate corresponds to its hilly, sheltered characteristics, with an oceanic tinge [54]. To speed up the transition to a circular economy in the Sibiu Depression's agri-food sector, we must promote an open attitude towards the subordination of the economy to the environment and the integration of production activities with the processing of agri-food products.

Our analysis of legislative factors includes the laws and regulations on the transition to a circular economy. We discovered that the rapid transposition of European legislation into national legislation on environmental protection and waste management, in addition to other circular economy initiatives, has had a positive impact on accelerating the transition to a circular economy.

### 3.2. The Relevance of the PESTEL Model Criteria for the Transition to the Circular Economy

Our diagnosis of the area using the PESTEL analysis model allowed us to identify specific elements of the local community, enabling the decision makers and stakeholders in agri-food development to better implement strategies to transition to a circular economy model. In this way, local rural development actors acquire the ability to better relate to concrete conditions and, consequently, gain a better capacity to design and implement policies and strategies.

The aspects that characterise the studied area are presented grouped into the six major analysis criteria according to the PESTEL model. Within these criteria, 19 sub-criteria were identified (3 belong to the political criterion, 4 to the economic criterion, 3 to the social criterion, 4 to the technological criterion, 3 to the average criterion, and 2 to the legislative criterion) relevant for the elaboration of the directions to be followed in order to accelerate the transition from the linear economic model to the circular model (Table 1). The relevance of these sub-criteria is presented in Table 1 and was obtained with the help of agri-food specialists. They were asked to assess, on a 5-level scale from 1 to 5 points, where 1 represents a not-significant assessment and 5 a very significant assessment, the importance of each sub-criteria for the development of strategies regarding the implementation of the circular economy in the agri-food sector.

**Table 1.** Relevance of PESTEL criteria for the transition to the circular economy.

| Field of Diagnosis | Subcriteria | Level of Impact on Future Strategies | | | | |
|---|---|---|---|---|---|---|
| | | 1 | 2 | 3 | 4 | 5 |
| **Political** | Creating an appropriate legislative framework at EU level | | | | | ■ |
| | Update national legislation | | | | | ■ |
| | Promoting local policies to support the process | | | | ■ | |
| **Economic** | Good representation of agri-food businesses | | | | ■ | |
| | Developed infrastructure for agri-food business development | | | | ■ | |
| | Promotion of elements of financial support for the process | | | | | ■ |
| | Skilled workforce | | | | ■ | |
| **Social** | 50% of the population of Sibiu County | | | | ■ | |
| | 16% of the population lives in rural areas | | | | ■ | |
| | High level of school education | | | | ■ | |
| **Technological** | Physical and ICT infrastructure well represented | | | | ■ | |
| | High level of innovation expenditure | | | | ■ | |
| | High number of agri-food enterprises | | | | ■ | |
| | Research and Development expenditure | | | | ■ | |
| **Environmental** | Environmental protection and pollution control policies | | | | | ■ |
| | Integration of production activities with agricultural processing activities | | | | ■ | |
| | Rapid transposition of European legislation into national legislation on environmental protection and waste management | | | | | ■ |
| **Legal** | Regulations on: circular economy | | | | | ■ |
| | Environmental protection: | | | | | ■ |

### 3.3. Results of Using the SWOT Model

Our results are relevant; however, for a better and more complete summary of the socio-economic characteristics of the Sibiu Depression microregion, we also used the SWOT analysis model. This allowed us to identify strengths and weaknesses, such as external opportunities or threats to an organization or territorial unit [54]. The information we obtained led us to the following conclusions regarding strengths and weaknesses, i.e., opportunities and threats to the sustainable development of rural tourism and agri-tourism within the SWOT matrix (Table 2).

**Table 2.** SWOT Analysis.

| | Strengths | | Weaknesses |
|---|---|---|---|
| 1 | The creation at EU level of a legislative framework dedicated to stimulating the transition to a circular econom. | 1 | Weak reaction of political forces to update national legislation. |
| 2 | Intense concern at EU level about the transition to circular farming. | 2 | Low resource productivity and recycling. |
| 3 | Much of what we consider today as waste in the agri-food sector can be processed and used. | 3 | Lack of support for policy makers and stakeholders to develop specific standards. |
| 4 | Support from local authorities to restore a balance between socio-economic interests and their impact on the environment. | 4 | Poor implementation of integrated waste management legislation. |
| 5 | Promoting integrated farming systems. | 5 | Few opportunities created to implement sustainable business models. |
| | Opportunities | | Threats |
| 1 | Production models based on the 10R philosophy are applicable to the agri-food sector. | 1 | Lack of policies and strategies towards integration of activities at farm/producer level. |
| 2 | Promote partnership and encourage innovation in redesigning the current supply chain (farm to fork). | 2 | The lack of measures to make the economy organic and to integrate production and processing activities. |
| 3 | The post-2020 funding framework offers relevant opportunities for resource efficiency and implementation of the circular economy. | 3 | Lack of funding for innovation and infrastructure in the transition to the circular economy. |
| 4 | The relatively high level of innovation expenditure and number of enterprises favored by the existence of the two industrial parks and touristic reputation. | 4 | Low public awareness of the need to adopt sustainable consumption patterns. |
| 5 | High capacity for education and information on the circular economy. | 5 | Lack of policy measures to improve the productivity and quality of agri-food production. |
| 6 | Development of smart agriculture. | | Predisposition towards over-consumption of resources. |

The information presented in our SWOT analysis leads us to the following conclusions:

- The important advantages of the circular development of the agri-food sector in the Sibiu Depression are the chance to use and process a large part of waste agri-food sector and the implication of the local authorities in balancing the socio-economic interests and their impact on the environment.
- The most dangerous weaknesses are the lack of interest from the political forces towards updating national legislation and a low recycling level, which connect with a few options created for the implementation of new sustainable business models.
- Taking into consideration the advantages, the implementation of a circular economy in the Sibiu Depression can apply the 10R-philosophy-based production models to the agri-food sector by encouraging innovation in redesigning the current supply chain and utilizing the high capacity for education and information.

### 3.4. Results of Using the DPSIR Model

We achieved the consolidation of strategic options for the agri-food sector's transition to a circular economy in the Sibiu Depression using the DPSIR model (Figure 3), which provides an integrated and dynamic view of the specific decision-making processes and reflects the relationships and cause-effect cycles between the 5 categories [55]. The use of the DPSIR model consisted of evaluating the current situation regarding the transition to the circular economy in the agri-food sector by considering a set of 5 criteria (driving forces, pressures, status, impact, and response) for which specific indicators were identified. They are grouped as follows: four from the category of driving forces—agro-food technologies, dynamics of agro-food enterprises, innovation expenses, research and development expenses; four from the category of pressures–soil degradation, water pollution, air pollution, use of alternative energy; four from the state category—qualified workforce, high level of education, quality of the agricultural environment, production and consumption models; four from the impact category–the integration of activities, the tendency to change production and consumption models, environmental protection, health and well-being; and three from the response category—the transposition of European legislation into national legislation, balance between economic and social interests, economic development strategies circular. The indicators were established in accordance with the sub-criteria identified within the PESTEL model. Three criteria were established for the evaluation of the indicators: relevance for the development of strategies, responsiveness, and data accessibility. The evaluation consisted of awarding points from 0–3 (0 = not useful, and 3 = very useful) during a focus group meeting attended by specialists in the field and factors of local responsibility. The activity consisted of drawing up some tables with the evaluation criteria, the score, and the indicators grouped by each of the 5 criteria of the model. Data centralization highlighted the usefulness of the indicators used.

In our DPSIR analysis, we focused on the R-factor, which is expected to provide relevant responses to interactions between the analysed parties: driving forces (agri-food consumption needs) and pressures (effects on the socio-economic, natural, and business environments), and the relationship between states (the state of the environment and natural resources) and impact (health and wellness). Responses to these interactions, taking into account our methodology, aim to accelerate the process of developing policies, programs, and strategies supported by local authorities that are focused on the circular development of the agri-food sector, the development of new sustainable business models within the agri-food sector, and education and information on the circular economy.

We synthesised this information and elaborated the following strategic options for the circular development of the agri-food sector in the Sibiu Depression:

- The transposition of European legislation on the circular economy must be a priority, and we must monitor the achievement of the objectives;
- We must implement policies, programs, and strategies for the circular development of the agri-food sector. Intervention must consider territorial specificity and be oriented towards the efficient use of resources and the promotion of sustainable businesses based on the 10 Rs philosophy. By doing so, we can create opportunities for the implementation of sustainable business models with the help of the post-2020 funding framework;
- We must leverage greater education and information capacities to raise public awareness regarding the need to adopt sustainable consumption patterns;
- We must create integrated farming systems by promoting partnerships and encouraging innovation, thereby redesigning the current supply chain (farm to fork).

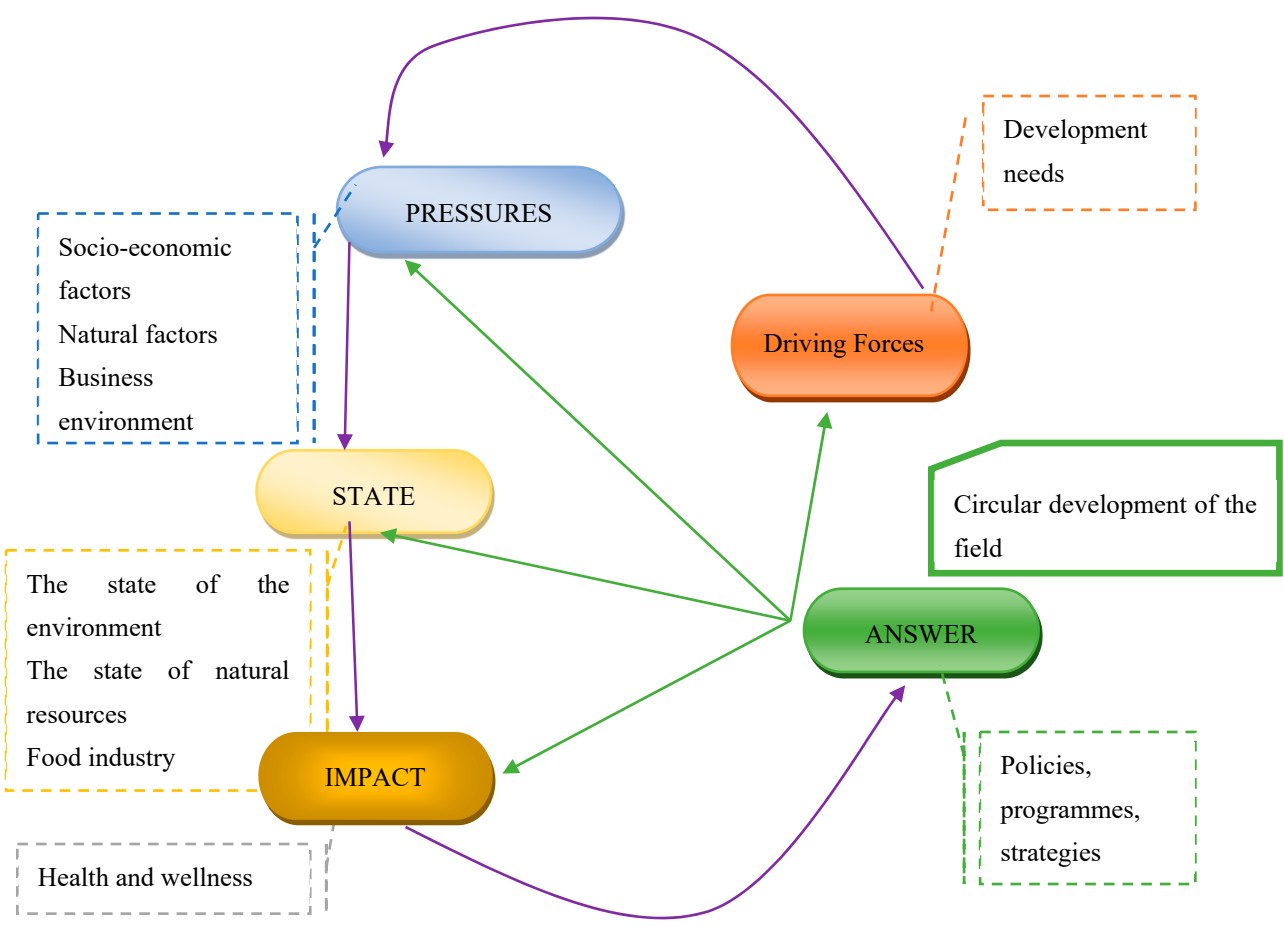

**Figure 3.** DPSIR conceptual framework on agri-food sector.

## 4. Conclusions

The diagnosis made in this study allows the responsible factors to better guide the process of transitioning to the circular economy because it captures relevantly the territorial specificity in relation to the addressed field. It represents a model that authorities can adopt.

Our elaboration of relevant strategic options for the circular development of the agri-food sector in the Sibiu Depression is based on a diagnosis of its component elements.

We carried out our diagnosis using a case study methodology, and we identified success factors and shortcomings regarding the circular development of the agri-food sector in the Sibiu Depression.

Our research methodology used strategic management methods and consisted of a secondary analysis of the relevant literature, identifying critical factors and successful initiatives, and applying the PESTEL and SWOT analysis models. Our results allowed us to configure the DPSIR model (driving forces, pressure, state, impact, answer) at a microregional level, commencing an integrated and dynamic transition from a linear to a circular economic model for the agri-food sector.

Our study highlights the existence of necessary prerequisites for the circular development of our study area's agri-food sector because of investments made in the tasks of obtaining raw agricultural materials (elements of intelligent technology), processing them (automation of production processes), and gathering educational materials and information on the circular economy.

For the agricultural sector at EU level, there are intense concerns about the transition to circular agriculture. The transition is focused on partnership and innovation, which creates an appropriate framework for sharing knowledge and experience towards the redesign of the current supply chain.

Our diagnosis of the area using PESTEL and SWOT analysis models identified specific community characteristics and enabled agri-food decision makers and stakeholders to better guide the elaboration and implementation of strategic options in the transition to the circular economy model.

We achieved a consolidation of strategic options for the agri-food sector's transition to a circular economy in the Sibiu Depression using the DPSIR model, which provides an integrated and dynamic view of the specific decision-making context. Thus, the legislative framework's role is dedicated to stimulating the transition to a circular economy and updating national legislation, including transposing European legislation directives, improving productivity to increase the quality of production, increasing positive synergies at the level of external environmental analysis criteria, and promoting new sustainable business models.

This work, completed with the development of strategic options for the circular development of the agro-food sector at the level of the Sibiu Depression, makes an important contribution to the transition process from the linear to the circular economic model. The research represents a model of strategic analysis at the territorial level for capturing the elements of specificity and developing strategic options for their sustainable exploitation by promoting the circular economy. This model can be replicated in different areas with specific characteristics. At the same time, the research directs the factors of local responsibility toward actions with an impact on the development of the circular economic model in the agri-food sector using strategic thinking and strategic management methods.

**Author Contributions:** Conceptualization, R.I. and A.Ș.; methodology, R.I.; validation, R.I., A.Ș. and P.I.; writing—review and editing, R.I. and A.Ș.; supervision, P.I.; project administration, R.I. All authors have read and agreed to the published version of the manuscript.

**Funding:** This research was funded by "Lucian Blaga" University of Sibiu and Hasso Plattner Foundation research grants LBUS-IRG-2020-06.

**Institutional Review Board Statement:** Not applicable.

**Informed Consent Statement:** Not applicable.

**Data Availability Statement:** Not applicable.

**Conflicts of Interest:** The authors declare no conflict of interest.

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
