# Peer review of "Strategic Thinking and Its Role in Accelerating the Transition from the Linear to the Circular Economic Model—Case Study of the Agri-Food Sector in the Sibiu Depression Microregion, Romania"

_sustainability, doi:10.3390/su15043109_

Round 1

Reviewer 1 Report (Previous Reviewer 3)

The revised paper is improved accordingly to the presented comments.

Author Response

Thank you very much to the reviewer. 

Reviewer 2 Report (New Reviewer)

My first substantive comment relates to the title. It suggests something generic but the paper focuses on one sector, in one region, of one country. The title must be modified to reflect the actual situation. One suggestion is referring to the paper as a case study.

My second substantive comment relates to the conclusion. You state that your diagnosis "enabled decision-makers and stakeholders to better guide...." This suggests that your methodology has already been applied by governments and others. It is not clear to me that this is the case.

Other comments:

The word "strategic" is used too often in the first few lines of the Intro. 

Lines 66 etc. I am surprised that your list of issues does not include climate change.

Lines 85-87. The circular economy may be a serious topic on the political agenda in Europe but this is not the case globally.

Line 95. What are the ten R's concept and 14.0 technology?

Lines 94-110. This is an odd list. Some items are generic  eg. minimizing the use of natural resources in production, while others are very specific eg.leasing. Consistency would be better or separating the list into two categories.

Line 159. What is "beech trout" and why is it relevant?

I do not think the authors can claim broad success regarding their premise when the study focuses on one sector in one region of one country.

Line 242. It is not clear how the authors defined and selected "good practices".

Lines 342-346. It is not clear how these indicators were selected. For example the cost of utilities is mentioned but not the cost of energy.

Line 359. What is meant by "the aim is to achieve synergy between the food industry and agricultural production"? Is the intention to reduce food waste?

I find this to be quite a complicated decision-making process involving several methodologies. Will it actually be effective in a bureaucracy?

Author Response

Thank you for reviewing our work and for your observations. We acknowledge that your suggestions contributed to improving the quality of our paper and your feedback is appreciated. Please see below the answers to your latest remarks!

  1. My first substantive comment relates to the title. It suggests something generic but the paper focuses on one sector, in one region, of one country. The title must be modified to reflect the actual situation. One suggestion is referring to the paper as a case study.

Thank you very much for your observation! Yes, we agree with the reviewer, and we made some changes in the article ‘title:

Strategic thinking and its role in accelerating the transition from the linear to the circular economic model. Case study of the agri-food sector in the Sibiu Depression microregion, Romania

  1. My second substantive comment relates to the conclusion. You state that your diagnosis "enabled decision-makers and stakeholders to better guide...." This suggests that your methodology has already been applied by governments and others. It is not clear to me that this is the case.

We applied the reviewer suggestion. We completed the conclusions with the following text:

“The diagnosis made in this study allows the responsible factors to better guide the process of transition to the circular economy because it captures relevantly the territorial specificity in relation to the addressed field. It represents a model that authorities can adopt.”

Other comments:

  1. The word "strategic" is used too often in the first few lines of the Intro.

We applied the reviewer suggestion. We reduced some number of this word as it can be seen in the manuscript.

  1. Lines 66 etc. I am surprised that your list of issues does not include climate change.

Thank you very much for the observation! We added in the manuscript clarifications, written in violet color. 

  1. Lines 85-87. The circular economy may be a serious topic on the political agenda in Europe but this is not the case globally.

Thank you very much for the observation! We added in the manuscript clarifications, written in violet color.

  1. Line 95. What are the ten R's concept and 14.0 technology?

It is about the Industry 4.0 (not 14.0) or I4.0 concept (transforming industrial production through digitalization and exploiting the potential of new technologies, a strategy to improve technology)

 The 10 R’s concept =Refuse, Rethink, Reduce, Reuse, Repair, Refurbish, Remanufacture, Repurpose, Recycle, and Recover

  1. Lines 94-110. This is an odd list. Some items are generic  eg. minimizing the use of natural resources in production, while others are very specific eg.leasing. Consistency would be better or separating the list into two categories.

Thank you very much for the observation! We made changes in the manuscript, written in violet color.

  1. Line 159. What is "beech trout" and why is it relevant?

– Pleurotus ostreatus Jacq (fungus species cultivated on an international scale)

  1. I do not think the authors can claim broad success regarding their premise when the study focuses on one sector in one region of one country.

We completed the manuscript with the text:

“Territorial specificity is relevant for the adoption of development directions, and once the general strategic framework is created through legislative measures developed at the European and national level, its approach in relation to the agri-food field highlights relevant options for transition to the circular economy.”

  1. Line 242. It is not clear how the authors defined and selected "good practices".

Thank you very much for the observation! We made changes in the manuscript, written in violet color.

  1. Lines 342-346. It is not clear how these indicators were selected. For example the cost of utilities is mentioned but not the cost of energy.

Thank you very much for the observation! We made changes in the manuscript, written in violet color.

  1. Line 359. What is meant by "the aim is to achieve synergy between the food industry and agricultural production"? Is the intention to reduce food waste?

Our answer is the following, and we completed the manuscript with the text:

“At the level of the Sibiu Depression, the aim is to achieve a synergy between the food industry and agricultural production, respectively a substantial added value as a result of the completion of the two areas and the intensification of specific activities in the conditions where the trend of these activities is an increasing one.”

  1. I find this to be quite a complicated decision-making process involving several methodologies. Will it actually be effective in a bureaucracy?

Our answer is the following:

The methodology leads to highlighting the elements of territorial specificity in relation to the field
agrifood. At the same time, it allows the development of relevant strategic options for accelerating the transition to the circular economy even if it is influenced by bureaucracy.

This manuscript is a resubmission of an earlier submission. The following is a list of the peer review reports and author responses from that submission.

Round 1

Reviewer 1 Report

Dear Authors,

Please consider the following comments and suggestions:

1. Use a proofreader for your English - there are many repetitions, typos, grammar errors etc.

2. Contract general information into more coherent and clear sentences, as for example lines 28-45

3. Figure 1 - it is unclear, I think there are concepts missing

4. Check Author's Guidelines for referencing and citations.

5. Divide the 'Results and discussions' section into 2-3 sub-topics according to the research objectives, so as to help readers find more easily information within the article.

6. Table 1 - Please explain the scores on the Likert scale. Who performed scoring? Based on what? 

7. The methodology used does not comply with the rigour of this journal. You just provide a subjective characterization of a region.

Yours faithfully,

Author Response

Response to Comments of Reviewer 1

Strategic thinking and its role in accelerating the transition from a linear to a circular economic model

Thank you once again for reviewing our work and for your observations. We acknowledge that your suggestions contributed to improving the quality of our paper and your feedback is appreciated. Please see below the answers to your latest remarks! We write with red our answer for reviewers and, we marked in text with red the updated text (what we inserted or what we changed).

  1. English language and style

( ) Extensive editing of English language and style required
(x) Moderate English changes required

( ) English language and style are fine/minor spell check required
( ) I don't feel qualified to judge about the English language and style

With all respect for the reviewer, at the beginning, before submitting the article we have undergone English language editing by MDPI service. The text has been checked for correct use of grammar and common technical terms and edited to a level suitable for reporting research in a scholarly journal. We took into consideration all suggestions and applied in our paper and then have uploaded on MDPI platform. On this stage, I upload together with the answers for reviewers also the English Editing Certificate (English-Editing-Certificate-53292.pdf).

  1. Contract general information into more coherent and clear sentences, as for example lines 28-45

Thank you very much for your observation! Yes, we agree with the reviewer, and we reorganized the specified part of the text in the manuscript:

“The general context in which humanity must establish relevant objectives and strategies for the transition to the circular economy is dominated by a summation of negative effects as a result of the linear economic model. This led to the "depletion of the natural capital on the planet considered our home" [2]; the decrease and degradation of natural resources, air and water pollution, respectively the decline of natural ecosystems [3,4]. So, the concerns of specialists regarding the acceleration of the transition to the circular economic model are justified and have been intensifying since the last decade. There is a unanimous recognition that the economy developed on "take, make, consume and dispose" is not sustainable, harms the health of the environment, and contributes to the decline of natural ecosystems [5-9]. A new orientation is needed, focused on the sustainability of ecosystems based on strategic thinking whose economic goal is to maintain as much as possible the value of products, materials, and resources and to significantly reduce the amount of waste. Thus, the circular economy manages to attract more and more followers who energize the approaches of specialists regarding the definition of the concept, far from reaching a unanimously accepted definition. Most specialists revolve around the following key concepts: sustainable development, the framework of the 4Rs (Reduce, Reuse, Recycle, Recover), the systemic approach (micro, meso, macro), the waste hierarchy [10]. The need to promote economic, social, environmental and technological elements in the process of development is also obvious [11]. The successful implementation of the circular economy concept needs political and legislative support, which is why it is becoming a ubiquitous topic on the global political agenda [12]. Next, we can see the generation of instruments that support the transition to the circular economy model, such as: the circular economy package [13], the law for the promotion of the circular economy in China [14] etc.”

  1. Figure 1 - it is unclear, I think there are concepts missing

Yes, we agree with the reviewer. We rethought and added new elements in the figure.

Figure 1. Sustainability of the economy based on the circular economy

            “The succinct image of what is presented is successively captured in figure 1 where ensuring the sustainability of the economy is attributed to the implementation of the circular economy by accelerating the transition to it and promoting its key elements at the level of the economic, social, environmental and technological pillars.”

  1. Check Author's Guidelines for referencing and citations

We reorganized and reedited all the references and citations in according with the Author's Guidelines.

  1. Divide the 'Results and discussions' section into 2-3 sub-topics according to the research objectives, so as to help readers find more easily information within the article.

Thank you very much for your observation! Yes, we agree with the reviewer, and we reorganized the specified part of the text in four sub-topics, as was modified in the manuscript:

3.1. Obtained results from using PESTEL diagnostic model

3.2. The relevance of the PESTEL model criteria for the transition to the circular economy

3.3. Results of using the SWOT model

3.4. Results of using the DPSIR model

  1. Table 1 - Please explain the scores on the Likert scale. Who performed scoring? Based on what? 

Thank you very much for this remark. We introduced more explanations in the manuscript:

“The aspects that characterize the studied area are presented grouped in the six major analysis criteria according to the PESTEL model. Within these criteria, 19 sub-criteria were identified (3 belong to the political criterion, 4 to the economic criterion, 3 to the social criterion, 4 to the technological criterion, 3 to the average criterion and 2 to the legislative criterion) relevant for the elaboration of the directions to be followed in order to accelerate the transition from the linear economic model to circular model (Table 1). The relevance of these sub-criteria is presented in table 1 and was obtained with the help of agri-food specialists. They were asked to assess on a 5-level scale from 1 to 5 points where 1 represents the not significant assessment and 5 the very significant assessment, the importance of each sub-criterion for the development of strategies regarding the implementation of the circular economy in the agri-food sector.”

  1. The methodology used does not comply with the rigour of this journal. You just provide a subjective characterization of a region.

With all respect for the reviewer, we have the following arguments, and we modified also in the manuscript:

“The research methodology used is recommended by the utility proven in numerous similar studies, such as: analysis of the area for new businesses in the raw materials market [38]; circular economy strategies, implementation and integration [39]; qualitative analysis of stakeholders for a regional biogas development [40]; integrated swot-pestel-ahp sustainability assessment model [41] etc.”

Reviewer 2 Report

1. The research gap needs to be appropriately addressed. 

2. Contribution of the paper is not clear.

3. Why is micro-region of the Sibiu Depression was investigated? Why is the agricultural sector was investigated? The paragraph for this reasoning needs to be modified with solid arguments.

4. What motivates the author(s)? How the analyze can be a helping hand for the countries (e.g. EU) in their pursuit of tackling linear economic?

5. The theoretical background of the article is at a very low level. There are no leading literature on the discussed issues.

6. Research methodology should be better presented - the methodology is described very generally eg. A secondary analysis of statistical data and relevant literature (reports, strategies, studies, monographs etc.) followed, which together with the previous information led to a realistic picture.”. How was a questionnaire built?

7. Conclusion needs to be conscious and informative.

Author Response

Response to Comments of Reviewer 2

Strategic thinking and its role in accelerating the transition from a linear to a circular economic model

  1. The research gap needs to be appropriately addressed
  2. Contribution of the paper is not clear.

Thank you for reviewing our work and for your observations. We acknowledge that your suggestions contributed to improving the quality of our paper and your feedback is appreciated. Please see below the answers to your latest remarks! We write with red our answer for reviewers and, we marked in text with red the updated text (what we inserted or what we changed).

We applied the reviewer suggestion. We completed the manuscript at conclusions:

“This work, completed with the development of strategic options for the circular development of the agro-food sector at the level of the Sibiu Depression, makes an important contribution to the transition process from the linear to the circular economic model. The research represents a model of strategic analysis at the territorial level for capturing the elements of specificity and developing strategic options for their sustainable exploitation by promoting the circular economy. This model can be replicated in different areas with specific characteristics. At the same time, the research directs the factors of local responsibility towards actions with an impact on the development of the circular economic model in the agri-food sector using strategic thinking and strategic management methods.”

3.Why is micro-region of the Sibiu Depression was investigated? Why is the agricultural sector was investigated? The paragraph for this reasoning needs to be modified with solid arguments.

Our answer is the following and we completed the manuscript with the text:

         “The motivation for choosing the Sibiu Depression microregion as the object of the case study in this research is based on its high habitat and geoproductive potential. The Sibiu Depression is located in Romania, more precisely in the southwest of the Transylvanian Hilly Depression and in the northern part of the Southern Carpathians, being polarized by the city of Sibiu - a strong urban center in continuous development. A municipality, two towns, 9 communes and 8 belonging villages are part of the Sibiu depression. The agricultural area totals 47495 ha of which 58.73% belongs to the categories of pastures and hayfields, and 40.01% belongs to the category of arable use, which highlights the favorability for animal breeding and the development of the food industry. The Sibiu Depression is located on the outskirts of a traditional area recognized for its tradition in animal husbandry, the promotion of traditions and the obtaining of food products, namely Mărginimea Sibiului. At the same time, the development of agri-food education in the city of Sibiu, both pre-university and university type, justifies the choice of this theme.”

  1. What motivates the author(s)? How the analyze can be a helping hand for the countries (e.g. EU) in their pursuit of tackling linear economic?

Our answer is the following:

“The Sibiu Depression is a micro-region with multiple development possibilities, including for the agri-food sector where the tradition of raising animals and processing agricultural products is recognized by numerous brands (Sibiu salami, telemeaua de Sibiu (Sibiu chees), etc.). This forces the continuation of the development process of the agri-food sector, but under the conditions of environmental protection through the implementation of the circular economy concept. The research presents the implications of strategic thinking, respectively of strategic management in the development of strategic options for accelerating the transition to the circular economic model. Their development required the development within the research of a model for collecting data and quantitative and qualitative information, for strategic diagnosis of the area to highlight specific characteristics, for combining internal and external characteristics, for an integrated and dynamic vision of the specific decision-making context, respectively highlighting cyclical cause-effect relationships. This model can be replicated in different areas with specific characteristics.”

  1. The theoretical background of the article is at a very low level. There are no leading literature on the discussed issues.

Thank you very much for the observation! We added in the manuscript the following clarification:

“The economic and social context is favorable to the development of the circular economy as can be seen from the interest of researchers, from scientific publications and not least from the legislative and financial support for the promotion of this concept. The circular economy in the agri-food sector is of particular importance due to the role it plays in the generation of welfare and social development, as well as in the balance of the environment [31]. Also, an analysis carried out by Hamam et al. regarding circular economy models in the agri-food sector highlights the need to implement cleaner production models [32]. In agriculture, the circular economy is seen as an economic model that respects the environment and offers emerging business opportunities [33].”

  1. Research methodology should be better presented - the methodology is described very generally eg. A secondary analysis of statistical data and relevant literature (reports, strategies, studies, monographs etc.) followed, which together with the previous information led to a realistic picture.”. How was a questionnaire built?

Thank you very much for the observation! We added in the manuscript the following clarification:

            “In order to achieve the goal, namely, to develop relevant strategic options to accelerate the transition process from the linear to the circular economic model, the research was carried out in three stages. The first stage consisted of collecting data and information useful for the research. Concretely, a questionnaire focused on the collection of data and quantitative information at the level of territorial administrative unit (UAT) and the secondary analysis of statistical data and relevant specialized literature was applied. These led to the shaping of a realistic picture and the identification of critical factors, respectively good practices. To bring more knowledge, participatory observation was also used, a qualitative method of collecting data and information that allowed combining data sets and obtaining answers to comparative questions [36]. The second stage is dedicated to the strategic diagnosis made at the level of the Sibiu Depression microregion by using the PESTEL and SWOT strategic analysis models. The use of the PESTEL analysis model assumed the grouping of the life framework of the Sibiu Depression into a set of 6 criteria (political, economic, social, technological, environmental and legislative), their analysis, the identification and understanding of the macroeconomic forces with an impact on the transition to the model circular economy in the agri-food sector. Specific characteristics of the community/area studied were identified and an important step was taken to develop relevant strategic options. In addition to the obtained information, the SWOT analysis model was used, with the help of which the specific internal and external characteristics of the studied community were combined. The combination of these characteristics outlined four quadrants to which certain strategic options correspond. The third stage is dedicated to the development of relevant strategic options for accelerating the transition from the linear to the circular economic model in the agri-food sector. For this, we proceeded to identify the factors and understand the relationship between them and the processes of implementing the circular economy using the DPSIR model. The application of the model led to the highlighting of the relationship between the "driving forces" and the political response [37] and to the establishment of an integrated and dynamic vision to accelerate the transition process to the circular economic model for the agri-food sector. Interactions were identified between the analyzed components such as those related to consumer need (driving forces) and its effects on the environment (pressures) under the impact of a certain production and consumption model. This interaction generates the need for change so that with the help of technologies the present situation (state) can be overcome, respectively the impact on the environment by obtaining the most relevant answer to the question of what must be done for the transition to the circular development of the agri-food sector.”

  1. Conclusion needs to be conscious and informative.

Thank you very much for the observation! We improve the conclusions as we mentioned above (point 2).

Reviewer 3 Report

The manuscript aims at applying the strategic thinking framework to the transition of the agri-food sector from the linear to the circular economy approach, considering a use case.

The research motivations are quite clearly described, but the methodology explanation is incomplete and its application is quite confusing, so the results are only partially justified.

I suggest carefully revising the manuscript's structure and the presentation of the results. Below are my comments:

·        The bullet points list on pages 2-3 (lines 77 to 103) should be re-edited since the present form is scarcely readable: where the sentences continue between points the content is quite unclear.

·        Please, add text lines commenting the figure 1

·        Add a very brief description of how you apply the PESTEL and SWOT analyses in section 2 (lines 211-212)

·        The DPSIR model, used to analyse the interactions, should be introduced in section 2; moreover, it is unclear how the presented results are obtained thanks to the application of the model: how do you derive the response? Which indicators are used? Please, explain the procedure.

·        The use case should be presented before the presentation of the results, justifying why did you choose the Region (lines 288 to 290 should be used for the Region agricultural vocation presentation)

·        How did you assign the “level of impact on future strategies” in table 1? Why a 5 levels scale is used?

·        The bullet points list on page 10 (lines 371 to 392) does not add any information to the results presented in table 2. It seems a repetition. Please add some comments.

·        Explain the manuscript's main contributions to the literature and to practitioners and local authorities approaching the transition of the agri-food sector to the circular economy approach. Explain how (or why) your results can be generalised and used in different contexts.

Author Response

Response to Comments of Reviewer 3

Strategic thinking and its role in accelerating the transition from a linear to a circular economic model

English language and style

( ) English very difficult to understand/incomprehensible
( ) Extensive editing of English language and style required
(x) Moderate English changes required
( ) English language and style are fine/minor spell check required
( ) I don't feel qualified to judge about the English language and style

With all respect for the reviewer, at the beginning, before submitting the article we have undergone English language editing by MDPI service. The text has been checked for correct use of grammar and common technical terms and edited to a level suitable for reporting research in a scholarly journal. We took into consideration all suggestions and applied in our paper and then have uploaded on MDPI platform. On this stage, I upload together with the answers for reviewers also the English Editing Certificate (English-Editing-Certificate-53292.pdf).

The manuscript aims at applying the strategic thinking framework to the transition of the agri-food sector from the linear to the circular economy approach, considering a use case.

The research motivations are quite clearly described, but the methodology explanation is incomplete and its application is quite confusing, so the results are only partially justified.

I suggest carefully revising the manuscript's structure and the presentation of the results. Below are my comments:

Thank you for reviewing our work and for your observations. We acknowledge that your suggestions contributed to improving the quality of our paper and your feedback is appreciated. Please see below the answers to your latest remarks! We write with red our answer for reviewers and, we marked in text with red the updated text (what we inserted or what we changed).

  1. The bullet points list on pages 2-3 (lines 77 to 103) should be re-edited since the present form is scarcely readable: where the sentences continue between points the content is quite unclear.

Thank you very much for your observation! Yes, we agree with the reviewer, and we reorganized the specified part of the text in the manuscript:

“Scientific forums also express support for promoting a circular economy through the conduct of specific research and the dissemination of results through publications in the form of case studies, reviews, scientific reports, and articles, converging towards the following:

  • An enterprise-level adoption of strategies aimed at achieving sustainable production based on the 10Rs concept and I4.0 technology [15];
  • Judicious resource allocation and the use of I4.0 in agriculture [16];
  • The valorization of waste (sludge) from drinking water treatment plants [17];
  • Analyzing the perceptions of the circular economy in Romanian SMEs in according with the six ReSOLVE framework actions, half being correlated in terms of value creation (regeneration, optimisation, and exchange) [18];
  • Promoting leasing as a sustainable business by recirculating products and ensuring economic performance [19];
  • Implementing the 3R principles (reduce, reuse, and recycle) and efficiently managing natural resources [20];
  • Increasing education levels [21];
  • Minimizing the use of natural resources in production processes [22,23];
  • Optimizing resource consumption by implementing waste management legislation [24];
  • Reusing products that still have operational functionality [25];
  • Extending the useful life of products [26].”

  1. Please, add text lines commenting the figure 1

Yes, we agree with the reviewer. We rethought and added new elements in the figure. 

 Figure 1. Sustainability of the economy based on the circular economy

            “The succinct image of what is presented is successively captured in figure 1 where ensuring the sustainability of the economy is attributed to the implementation of the circular economy by accelerating the transition to it and promoting its key elements at the level of the economic, social, environmental and technological pillars.”

  1. Add a very brief description of how you apply the PESTEL and SWOT analyses in section 2 (lines 211-212)

Thank you very much for the observation! We added in the manuscript the following clarification:

“The use of the PESTEL analysis model assumed the grouping of the life framework of the Sibiu Depression into a set of 6 criteria (political, economic, social, technological, environmental and legislative), their analysis, the identification and understanding of the macroeconomic forces with an impact on the transition to the model circular economy in the agri-food sector. Specific characteristics of the community/area studied were identified and an important step was taken to develop relevant strategic options. In addition to the obtained information, the SWOT analysis model was used, with the help of which the specific internal and external characteristics of the studied community were combined. The combination of these characteristics outlined four quadrants to which certain strategic options correspond.”

  1. The DPSIR model, used to analyse the interactions, should be introduced in section 2; moreover, it is unclear how the presented results are obtained thanks to the application of the model: how do you derive the response? Which indicators are used? Please, explain the procedure.

Our answer is the following and we completed the manuscript with the text:

3.4. Results of using the DPSIR model

            We achieved the consolidation of strategic options for the agri-food sector's transition to a circular economy in the Sibiu Depression using the DPSIR model (Figure 3), which provides an integrated and dynamic view of the specific decision-making processes and reflects the relationships cause-effect cycles between the 5 categories [53]. The use of the DPSIR model consisted in evaluating the current situation regarding the transition to the circular economy in the agri-food sector by considering a set of 5 criteria (driving forces, pressures, status, impact, and response) for which specific indicators were identified. They are grouped as follows: 4 from the category of driving forces – agro-food technologies, dynamics of agro-food enterprises, innovation expenses, research and development expenses; 4 from the category of pressures – soil degradation, water pollution, air pollution, use of alternative energy; 4 from the state category – qualified workforce, high level of education, quality of the agricultural environment, production and consumption models; 4 from the impact category – the integration of activities, the tendency to change production and consumption models, environmental protection, health and well-being and 3 from the response category – the transposition of European legislation into national legislation, balance between economic and social interests, economic development strategies circular. The indicators were established in accordance with the sub-criteria identified within the PESTEL model. Three criteria were established for the evaluation of the indicators: relevance for the development of strategies, responsiveness, and data accessibility. The evaluation consisted of awarding points from 0-3 (0 = not useful, and 3 = very useful) during a focus-group meeting attended by specialists in the field and factors of local responsibility. The activity consisted of drawing up some tables with the evaluation criteria, the score and the indicators grouped on each of the 5 criteria of the model. Data centralization highlighted the usefulness of the indicators used.”

  1. The use case should be presented before the presentation of the results, justifying why did you choose the Region (lines 288 to 290 should be used for the Region agricultural vocation presentation)

Our answer is the following and we completed the manuscript with the text:

“The Sibiu Depression is a micro-region with multiple development possibilities, including for the agri-food sector where the tradition of raising animals and processing agricultural products is recognized by numerous brands (Sibiu salami, telemeaua de Sibiu (Sibiu chees), etc.). This forces the continuation of the development process of the agri-food sector, but under the conditions of environmental protection through the implementation of the circular economy concept. The research presents the implications of strategic thinking, respectively of strategic management in the development of strategic options for accelerating the transition to the circular economic model. Their development required the development within the research of a model for collecting data and quantitative and qualitative information, for strategic diagnosis of the area to highlight specific characteristics, for combining internal and external characteristics, for an integrated and dynamic vision of the specific decision-making context, respectively highlighting cyclical cause-effect relationships. This model can be replicated in different areas with specific characteristics.”

  1. How did you assign the “level of impact on future strategies” in table 1? Why a 5 levels scale is used?

“The aspects that characterize the studied area are presented grouped in the six major analysis criteria according to the PESTEL model. Within these criteria, 19 sub-criteria were identified (3 belong to the political criterion, 4 to the economic criterion, 3 to the social criterion, 4 to the technological criterion, 3 to the average criterion and 2 to the legislative criterion) relevant for the elaboration of the directions to be followed in order to accelerate the transition from the linear economic model to circular model (Table 1). The relevance of these sub-criteria is presented in Table 1 and was obtained with the help of agri-food specialists. They were asked to assess on a 5-level scale from 1 to 5 points where 1 represents the not significant assessment and 5 the very significant assessment, the importance of each sub-criterion for the development of strategies regarding the implementation of the circular economy in the agri-food sector.”

  1. The bullet points list on page 10 (lines 371 to 392) does not add any information to the results presented in table 2. It seems a repetition. Please add some comments.

Thank you very much to the reviewer. We rephrased the conclusions of the SWOT analysis:

 “The information presented in our SWOT analysis leads us to the following conclusions:

  • The important advantages of the circular development of the agri-food sector in the Sibiu Depression are the chance to use and process a large part of waste agri-food sector and the implication of the local authorities in balancing the socio-economic interests and their impact on the environment.
  • The most dangerous weaknesses are lack of interest from the political forces towards updating national legislation and a low recycling level connecting with a few options created for the implementation of new sustainable business models.
  • Taking into consideration the advantages, the implementation of a circular economy in the Sibiu Depression can apply the 10R-philosophy-based production models to the agri-food sector by encouraging innovation in redesigning the current supply chain and utilizing the high capacity for education and information.”
  1. Explain the manuscript's main contributions to the literature and to practitioners and local authorities approaching the transition of the agri-food sector to the circular economy approach. Explain how (or why) your results can be generalised and used in different contexts.

Our answer is:

“This work, completed with the development of strategic options for the circular development of the agro-food sector at the level of the Sibiu Depression, makes an important contribution to the transition process from the linear to the circular economic model. The research represents a model of strategic analysis at the territorial level for capturing the elements of specificity and developing strategic options for their sustainable exploitation by promoting the circular economy. This model can be replicated in different areas with specific characteristics. At the same time, the research directs the factors of local responsibility towards actions with an impact on the development of the circular economic model in the agri-food sector using strategic thinking and strategic management methods.”
